# Effect of Biogenic Amine-Degrading *Lactobacillus* on the Biogenic Amines and Quality in Fermented Lamb Jerky

**DOI:** 10.3390/foods11142057

**Published:** 2022-07-12

**Authors:** Xueying Sun, Erke Sun, Lina Sun, Lin Su, Ye Jin, Lina Ren, Lihua Zhao

**Affiliations:** Department of Food Science, College of Food Science and Engineering, Inner Mongolia Agricultural University, Hohhot 010018, China; m15774711156@163.com (X.S.); sun15848107303@163.com (E.S.); beautysunlina@126.com (L.S.); sulin820911@163.com (L.S.); jinyeyc@sohu.com (Y.J.); renlina0267@163.com (L.R.)

**Keywords:** fermented lamb jerky, bioamine-degrading, lactic acid bacteria, quality

## Abstract

This study compares five types of lamb jerky, namely, CO (without starter culture), PL-4 (with producing putrescine, cadaverine, histamine, and tyramine), BL4-8 (degrading putrescine, cadaverine, histamine, and tyramine), CL4-3 (degrading putrescine and tyramine), and X3-2B (degrading histamine and tyramine). A study was performed to examine the effects of starter culture on the physical–chemical quality, flavor, and biogenic amines (BAs) during fermentation and ripening. At the end of fermentation, the pH value of the BL4-8 group (4.75) was significantly lower than that of other groups (*p* < 0.05). After high-temperature roasting, the water activity (0.55), water content (22.6%), nitrite residue (0.41 mg/kg), and TBARS value (0.27 mg/100 g) of the X3-2B group were significantly lower than those of other groups (*p* < 0.05). The findings show that adding starter BL4-8, CL4-3, and X3-2B can increase the variety and content of flavor in the product. The levels of histamine, putrescine, and tyramine were significantly lower in the BL4-8, CL4-3, and X3-2B groups than in CO and PL-4 groups. This study shows that BL4-8, CL4-3, and X3-2B are potential starters for fermented meat products.

## 1. Introduction

Fermented jerky is a fermented meat product that is attractive in color, safe, unique in flavor, has a long shelf life, and is welcomed and loved by the majority of consumers of Inner Mongolia, Liaoning, Sichuan, Henan, and elsewhere in China [1,2]. The survey shows that 88.0% of consumers have eaten leisure jerky products [3]. During the process of salting, fermenting, drying, maturing, or roasting under natural or artificial control conditions with the help of microorganisms, the jerky undergoes a series of biochemical reactions and physical changes. In terms of nutritional value, lamb products provide food low in calories and fat, and high in protein content. It is a high-quality raw material for meat processing and has broad prospects in development and application. Jerky made using traditional processing techniques is not controllable due to natural conditions, resulting in poor quality: for example, high concentrations of harmful amines [4].

Biogenic amines (BAs) are generally metabolized from some amino acids due to spoilage bacteria with amino acid decarboxylase activity in meat products [5]. From the perspective of toxicology, histamine (HIS) and tyramine (TYR) are regarded as the most significant biogenic amines because consumption of a high content of them can lead to hypertension, headaches, palpitations, and vomiting in some people, according to the European Food Safety Authority (EFSA). Putrescine (PUT) and cadaverine (CAD) are much less harmful than HIS and TYR, but they react with nitrite to produce carcinogens, such as nitrosamine substances as a potential precursor [6]. Lu et al. found that the contents of tyramine, histamine, cadaverine, and putrescine were 4.96–771.52 mg/kg, 0.1–101.34 mg/kg, 3.98–1435.24 mg/kg, and 0.1–449.98 mg/kg, respectively, in traditional sausages in different regions of China, and the contents of biogenic amines in some samples were over the estimated safe levels of acute exposure (100 mg/kg) [7]. As a result, consumers have been paying close attention to the concentration of various biogenic amines in fermented lamb products.

Because meat and meat products are nutritious, delicious, and have a desirable texture, they are a key part of many people’s diet. It is important to understand the composition and dynamics of the microbe population as well as the interactions between them during the processing and storage of meat and meat products, which are essential for ensuring quality and safety. The role of functional microbes in meat and meat products has recently attracted considerable attention as the methods of microbial analysis have advanced. Zhang et al. found that adding *Lactobacillus Plantarum* and *Lactobacillus salivarius* into traditional smoked horse meat intestines resulted in the degradation of BAs due to the amine oxidase from the beginning [8]. Gardini et al. found that *Staphylococcus xylosus* S81 can influence the activity of microbial amino oxidases, thereby influencing the amount of histamine during ripening in sausage production [9]. Therefore, inoculation with a starter that has biogenic amine-degrading abilities promotes the quality of fermented meat products. Additionally, the addition of starter culture can reduce the pH value and produce bacteriocin to inhibit the growth of spoilage bacteria and pathogenic bacteria, and improve the color, playing an important role in the formation of the flavor of the product [10,11,12].

This study investigates the effects of selected starter cultures with excellent functional characteristics of degrading biogenic amines through a study of fermented lamb jerky, mainly evaluating the biogenic amine formation, volatile flavor substances, and physical and chemical indexes.

## 2. Materials and Methods

### 2.1. Materials

Lamb and additives were purchased from a local market. Lyophilized native starters, including *Lactobacillus*
*plantarum* PL-4, BL4-8, CL4-3, and X3-2B, were provided by the Meat Laboratory (College of Food Science and Engineering, Inner Mongolia Agricultural University, Hohhot, China).

### 2.2. Preparation of Fermented Lamb Jerky

The following five groups were manufactured with the same process formula. Lamb jerky inoculated with single *L. plantarum* PL-4, BL4-8, CL4-3, and X3-2B were noted as the PL-4, BL4-8, CL4-3, and X3-2B groups, respectively, and those without commercial starter cultures were marked as the CO group. The starter cultures were washed three times with the same amount of sterile saline solution based on the final level of 10^7^ CFU/g meat. First, the fascia and fat were removed from the fresh lamb. The meat was divided into large cubes along the direction of the meat fiber and cleaned with water. The lean meat was cut into strips (4 × 2 × 1 cm) of uniform size. Second, sugar (0.5%), glucose (0.1%), NaCl (2.5%), sodium nitrite (0.0002%), white pepper powder (0.5%), and ginger powder (0.5%) were added to the meat strips. The meat strips were inoculated with the starter by injection method and marinated for 12 h. After marinating, the strips were fermented for 36 h at 90% RH and 33 °C. The low-temperature roasting of jerky lasted for 2 h at 70–90 °C, and then the high-temperature roasting of jerky lasted for 1 h at 150 °C. Each group of the samples was collected during the process to determine the indicators (Figure 1).

### 2.3. pH, Water Activity, Color, TBARS, Nitrite, and Texture

An electronic pH meter (Mettler Toledo, Shanghai, China) and the LabMaster-a_w_ (Novasina AG, Schwyz, Switzerland) were used to measure pH and a_w_. Fully automatic colorimeters were used to observe the color of lamb jerky (Oik TC-P2A, Beijing, China). e value, as the comprehensive value of meat color, is calculated by the following formula: e = a*/L + a*/b* [13]. The texture was measured using a mass structure instrument (TA-XTplus, Shanghai, China) following the method of Luo et al. [14]. The thiobarbituric acid (TBARS, mg/100 g) and residue nitrite (mg/kg) were quantified according to the method of Ma, Nan, and Dai [15] and GB5009.33-2016 in China [16].

### 2.4. Volatile Flavor

A 5 g sample was chopped and put into a 20 mL headspace bottle, and 1 μL of internal standard dimethyltriheptanone (accurately weighed to 16.8 mg of dimethyltriheptanone and dissolved in 100 mL of methanol) was added. The aged SPME extraction head was absorbed in a 60 °C water bath for 30 min, inserted it into the gas chromatography sample inlet after adsorption, and desorbed it at 250 °C for 3 min. Gas chromatography conditions were as follows: capillary column DB-5 (30 m × 0.25 mm × 0.25 μm), helium carrier gas flow rate of 1 mL/min, injection port, and interface temperature of 250 °C, with no split mode for the injection mode. Quantitative analysis was performed as follows: the internal standard method was used for quantitative analysis. The content of each component in the sample was calculated by the peak area and the relative response value of the internal standard substance and each component.

Odor activity values (OAVs) indicate the contribution of each volatile compound to overall odor [17].The OAV is calculated by dividing each compound’s content by its odor threshold value in water [18].

### 2.5. Electronic Nose

An electronic nose detector (PEN3, Airsense, Germany) comprising an array of 10 sensors (W1C, W5S, W3C, W6S, W5C, W1S, W1W, W2S, W2W, and W3S), was used to discriminate odor patterns of different aroma models. The test conditions were as follows: 5 g of chopped fermented lamb jerky was placed in a 15 mL septa-sealed screw cap vial and equilibrated for 40 min at 60 °C. The aroma headspace above the sample was then introduced into the E-nose equipment at 300 μL/min, with a sampling interval time of l s and a cleaning time of 90 s, and sensors were exposed to the volatiles for 200 s as recommended, with a zero-point trimming time of 1 s. Each analysis was performed in triplicate.

### 2.6. Biogenic Amines

The determination of BAs was carried out by extraction and derivatization using the methods of Sun et al. [19]. Five grams of jerky was extracted with trichloroacetic acid (5%), the fat was removed with hexane, and the purification was performed using a 1:1 solution of chloroform and n-butanol, and then the sample was derivatized by dansulfonyl chloride. A 10 µL sample of each sample’s filtrate was injected into a chromatographic column (ZORBAX SB-C18: 250 mm × 4.6 mm, 5 µL particle size, Agilent) after derivatization and filtering with a 0.22 μm filter, and then high-performance liquid chromatography was performed (HPLC, Agilent1260, Agilent, Santa Clara, CA, USA). The separated BAs were identified and compared with the retention times of known standards (1,7-diaminoheptane). The calculated lines of regression were used to compute the amount of analytes in the samples by interpolation using an internal standard method. The calibration curves, i.e., peak area divided by internal standard versus concentration, were linear in a concentration range between 1 and 50 mg/L. The limit of detection and limit of quantification indicated that the sensitivity was high and can satisfy the needs of sample determination. The final content of BAs was expressed as mg/kg on a fresh matter basis.

### 2.7. Statistical Analysis

SPSS 26.0 (IBM, Chicago, IL, USA) was used to analyze the data. One-way ANOVA was used to determine the significance of the data. The graphs in the article were made with Origin 2018.

## 3. Results and Discussion

### 3.1. pH, a_w_, and Moisture Content

The pH, a_w_, and moisture content during the processing are shown in Table 1. The pH values in the CO, PL-4, BL4-8, CL4-3, and X3-2B groups decreased to 5.22, 5.18, 4.75, 4.95, and 5.11, respectively, after fermentation (Table 1). The increase in pH values in all groups after low-temperature roasting and high-temperature roasting could be due to alkaline substances produced by protein decomposition, such as proteolytic amines, which were induced by decarboxylase and bacterial proteases. BAs are synthesized by decarboxylase of free amino acids, and bacterial proteases induce proteolytic degradation, generating peptides, amino acids, and amines which have a buffering effect on the organic acids produced by lactic acid bacteria during fermentation [20,21].

The mean values of the initial water activity (a_w_) of the five groups were over 0.9 (Table 1). The a_w_ of the four experimental groups decreased significantly faster than that of the CO group (*p* < 0.05) after high-temperature roasting, which might be caused by the rapid decrease in the pH of the PL-4, BL4-8, CL4-3, and X3-2B groups, and approximation to the isoelectric point of the protein, leading to a decrease in the ability of the protein to bind water [22]. After high-temperature roasting, the a_w_ in five groups decreased to below 0.6, the moisture content from over 70% to 20.30%, 23.00%, 23.30%, and 22.60% of the CO, PL-4, BL4-8, CL4-3, and X3-2B groups, respectively (Table 1). The low a_w_ and moisture content can improve the safety of fermented lamb jerky. This article also shows the same results as Wang et al.’s study of strains on fermented sausage, which used pH and water activity as tests to determine when the starter was added [23].

### 3.2. Color and Texture

Lightness increased significantly (*p* < 0.05) in the five groups from the initial value to the value at the end of roasting (Table 2). Redness increased rapidly for all groups and the redness of the PL-4 was higher than that of the other groups during high-temperature roasting. It may be that the nitrite reaction is converted into NO, which then reacts with myoglobin in the meat to form nitromyoglobin, giving a stable red color [24]. In this study, the e value was introduced as the main parameter to judge the color of fermented lamb jerky. After salting, the e values of the BL4-8, CL4-3, and X3-2B groups were significantly higher than that of the CO group (*p* < 0.05). After roasting, the e values of the four groups with the starter were significantly higher than that of the CO group (*p* < 0.05), which was due to the increase in roasting temperature; the b* value and L* value also increased, resulting in the overall decrease in the e value. After high-temperature roasting, the e value of PL-4 was higher than that of the other groups and it can be seen that the starter PL-4, BL4-8, CL4-3, and X3-2B can improve the color of fermented lamb jerky.

The texture is an index that directly reflects the sensory quality of products. It can be seen from Table 3 that the hardness and chewiness of the CO group after high-temperature roasting were significantly higher than they were for the other four groups (*p* < 0.05). Additionally, the chewiness of the CL4-3 group was better than that of the other groups. The elasticity, cohesion, and recovery of the four starter groups were significantly higher than they were for the CO group (*p* < 0.05), which may be due to the reduction in pH value after fermentation and the denaturation of muscle protein to form gelatinous tissue, which improved the hardness and elasticity of the fermented dried mutton [25]. The above data shows that the starter can significantly improve the texture characteristics of fermented lamb jerky and thereby effectively improve the sensory quality of the product.

### 3.3. Residue Nitrite and TBARS

The evolution of the residue nitrite and TBARS of fermented lamb jerky is shown in Figure 2. Under weak acid conditions, the nitrite was diazotized using p-Aminobenzene Sulfonic Acid, followed by the addition of naphthalene ethylenediamine hydrochloride to produce purple dye. According to the color of the solution, nitrite content was determined. The proper addition of nitrite in fermented meat products can promote color development, but excessive addition or residue will form carcinogen nitrosamine with amines [26]. It can be seen in Figure 2A that after salting and fermentation, the residue nitrite content in the four groups of starters was significantly lower than that in the CO group (*p* < 0.05). It may be that the starter was degraded by nitrite reductase and its acid production; the residue nitrite was gradually reduced [27]. After low-temperature roasting, the nitrite content in the four groups of starters increased slightly but was significantly lower than it was in the CO group (*p* < 0.05). After high-temperature roasting, the residue nitrite in the X3-2B group was the lowest at 0.41 mg/kg. According to the above data, the starter culture PL-4, BL4-8, CL4-3, and X3-2B can reduce nitrite.

The TBARS value is an important index to evaluate the fat oxidation of meat products. Excessive fat oxidation will affect the flavor and sensory quality of meat products [28]. It can be seen in Figure 2B that, during the processing of fermented lamb jerky, the TBARS value showed an overall upward trend. After salting, the TBARS value of each group was 0–0.1, and the degree of oxidation was not obvious. After the fermentation, the TBARS value of each group began to rise. It may be that the oxidase in the raw meat tissue and the oxidative decomposition enzyme produced by microorganisms led to the decomposition and oxidation in each group of fermented lamb jerky, which increased the TBARS value [29]. After high-temperature roasting, the TBARS value of each group reached the maximum, and the TBARS value of the test group was significantly lower than that of the CO group (*p* < 0.05). This may be due to the oxidation of light and air in the environment and the increase in the lipid oxidation rate with the increase in temperature, which promoted the production of free amino acids and the decomposition of hydrogen peroxide [30]. The above data show that the four kinds of added starter could effectively improve the oxidation degree of fermented lamb jerky and prevent the excessive oxidation from affecting the flavor and edible quality of the product.

### 3.4. Electronic Nose Analysis

The volatility profile of samples subjected to the five groups was also performed by a PEN3 E-nose. The original response data generated by 10 sensors of an E-nose were collected and transformed into radar graphs, as shown in Figure 3. Generally, the similar shape of these radar graphs implies some similarities among these samples from different stages of processes [31]. The response values of the 10 sensors were positive, as can be seen from Appendix A. The response values of W1S (sensitive to methyl) and W1W (sensitive to sulfides) sensors increased (*p* < 0.05), which suggested that methyl compounds and sulfides were generated in five groups jerky. In addition, the response values of W2W (sensitive to aromatics and organic sulfur compounds), W2S (sensitive to alcohols, aldehydes and ketones), and W3S (sensitive to long-chain alkanes) sensors increased (*p* < 0.05), which suggests that organic sulfides compounds, alcohols, aldehydes, ketones and long-chain alkanes were generated in CL-4 group. Based on the above fingerprint analysis, the main compounds that distinguished the processed fermented dried mutton from each other were alcohol, aldehyde, ketone, sulfide, and other compounds. As a result of fermentation, the levels of alcohols, aldehydes, and ketones increased in each group, in which the microbes produced enzymes through growth and reproduction, after high-temperature roasting, the sulfide response strength of each group was greater, which may be because the Maillard reaction is promoted during processing [32].

As shown in Figure 4, the two-dimensional PCA map of each group during the processing of fermented lamb jerky was drawn using the response intensity values of 10 sensors of the electronic nose. The total contribution rates of the two cumulative variances (PC1 and PC2) during the processing were 90.8%, 92.2%, 88.9%, and 83%, respectively, indicating that the two principal components contained most of the odor information of the sample [33]. The collected data points of each group were scattered in different areas, indicating that the odor components of fermented dried mutton in each group were well-separated and different in the processing. During the processing of fermented jerky, the odors of the four starter groups and the CO group were dispersed, indicating that there were differences between the starter group and the CO group. The smell of fermented jerky with the starter was unique, and the PCA diagram of the odor response value of each group of fermented jerky was consistent with the expression information of radar fingerprint, indicating that the electronic nose combined with PCA could separate the five groups of jerky during the processing. Therefore, the flavor substances in the process of fermented jerky can be accurately determined and analyzed by GC-MS.

### 3.5. Volatile Flavor Analysis

To better understand the composition of flavor during the processing of fermented lamb jerky, we carried out a heat map analysis on the volatile flavor substances in each group. The results are shown in Figure 5. The *x*-axis represents the group, and the *y*-axis represents 31 flavor substances detected in each group, including 6 aldehydes, 2 acids, 4 ketones, 5 alcohols, 3 esters, and 10 olefins. The compounds were generated by carbohydrate metabolism, lipid oxidation, amino acid catabolism, bacterial esterification, and spices [28]. Based on the gradual change in color, the content changes in different flavor substances in each group of fermented jerky during the processing were identified. According to the cluster analysis of the heat map, it can be seen that there were obvious differences in the flavor substances of the four stages in each group.

The aldehyde content of fermented jerky in each group during the processing process was higher. Aldehydes are the main flavor compounds in fermented mutton products, which have a low threshold and create an oily and fruity flavor [34]. They mainly come from the oxidation of unsaturated fatty acids, such as oleic acid and linoleic acid and the degradation of amino acids [35]. After fermentation, hexanal (3.62 μg/g), with a green flavor, was detected in the BL4-8 group. After low-temperature roasting, heptanaldehyde (1.20 μg/g), with a barbecue flavor, and glutaraldehyde (2.59 μg/g), with a malt flavor, were detected in the PL-4 group and the X3-2B group, respectively. After high-temperature roasting, fat-flavored aldehyde (1.36 μg/g) from the BL4-8 group, fresh benzaldehyde (6.05 μg/g), and octanal (2.22 μg/g) from the CL4-3 group were found. Given the low threshold and the rapid generation of the lipid oxidation rate, these substances made a large contribution to the overall flavor of the product [36].

After high-temperature roasting, 3-hydroxy-2-butanone (12.54 μg/g) was detected in the BL4-8 group with a strong cream and fat aroma and a pleasant milk aroma, and 2,3-ocdione (22.14 μg/g) was detected in CL4-3 group, with a sweet cream aroma. Ketones are produced by the oxidation of unsaturated fatty acids [37]. Ketones contribute to the production of cream aroma and contribute to the overall flavor of fermented lamb jerky through the microbial-oxidation reaction and methyl ketone production activities or lipid oxidation [38].

Both 2,3-butanediol (3.57 μg/g) and α-terpineol (1.95 μg/g) were detected in the BL4-8 group, and 2-Tzol (0.35 μg/g,), which was detected in the PL-4 group, had camphor odor. The detection of alcohols may have come from the secondary hydroperoxide decomposition of n-3 and n-6 fatty acids, as well as reduced sugars, amino acids, and base compounds [39]. A high threshold for saturated alcohols, such as ethanol at 100,000 μg/g and positive hexanol at 2500 μg/g, had a lower effect on the overall flavor [40]. Like linalool, α-terpineol, and 2-Tzol, these compounds have relatively low flavor thresholds and also contribute to the overall flavor of fermented jerky [41].

After high-temperature roasting, methyl hexatate (7.24 μg/g) was detected in the BL4-8 group with a pleasant smell, and methyl butyrate (5.22 μg/g) and methyl potanate (4.27 μg/g) were detected in CL4-3 group with an apple aroma and a fruit aroma, respectively. Ester is considered a source of fruit and caramel flavor, often through the esterification reactions of acids and alcohols and related to the esterase activity of microorganisms [42].

To sum up, the contents of flavor substances varied in each group during the processing of fermented jerky, and the type and content of flavor substances increased slightly after high-temperature roasting. From the above analysis, it can be seen that not only does the formation of flavor substances come from the automatic oxidation and degradation of fat, but also that the starter PL-4, BL4-8, CL4-3, and X3-2B had a great contribution to the formation of flavor substances, such as acetic acid, acetaldehyde, 2,3-butanedione, 3-hydroxy-2-butanone, and 2,3-butanediol, which are usually compounds produced by lactic acid bacteria through pyruvic acid metabolism, thus giving the fermented jerky a unique flavor [43].

Besides volatile compounds, flavor perception is also influenced by threshold values of volatile compounds, and these two factors are combined in the OAV index in order to evaluate volatile compounds’ contribution. There is a general acceptance that compounds with an OAV > 1 contribute significantly to overall odor [18]. A list of compounds with an OAV > 1 and their odor descriptions can be found in Table 4. There were 16 major volatile compounds with OAV > 1, thought to be the main contributors to the fermented odor of jerky. Particularly, hexanal, octanal, nonanal, 1-octene-3-ol, methyl butyrate, and methyl octanoate greatly contributed to the overall odor of fermented lamb jerky. The compound with the highest OAV (57.65–652.35) in all jerky was octanal, which contributed orange and honey odors to the jerky [44].

### 3.6. Biogenic Amines Analysis

BAs were accumulated in fermented lamb jerky during processing (Figure 6). Tryptamine (TRY) is a kind of monoamine alkaloid, which is formed by the decarboxylation of tryptophan and then oxidized by amine oxidase. As shown in Figure 6A, the TRY content of the PL-4, CL4-3, and X3-2B groups after fermentation was 7.14 mg/kg, 6.67 mg/kg, and 6.62 mg/kg, respectively, which are significantly lower than that of the CO group (71.12 mg/kg) (*p* < 0.05). After low-temperature roasting, the TRY content of the X3-2B group was 11.54 mg/kg, significantly lower than that of the other groups (*p* < 0.05). After high-temperature roasting, besides the CL4-3 group, the TRY content in other groups was significantly higher than that in the CO group (*p* < 0.05).

As shown in Figure 6B, after fermentation, the putrescine (PUT) content of the BL4-8, CL4-3, and X3-2B groups was significantly lower than that of the CO group by 5.10 mg/kg (*p* < 0.05). After low-temperature roasting, the PUT content accumulation in the PL-4 group reached a maximum of 11.06 mg/kg, which was significantly higher than that in the CO group, i.e., 7.15 mg/kg (*p* < 0.05). After high-temperature roasting, the PUT content in each group was consistent with that after fermentation and low-temperature roasting. The PUT content in the PL-4 group was significantly higher than that in other groups during the whole processing of fermented lamb jerky (*p* < 0.05), which may be because starter PL-4 promoted the formation of PUT in fermented lamb jerky, resulting in the accumulation of PUT content, while the PUT content in the BL4-8, CL4-3, and X3-2B groups were significantly lower than that in the CO group during the whole processing (*p* < 0.05), indicating that starter BL4-8, CL4-3, and X3-2B can reduce the PUT content in fermented lamb jerky. Bover-Cid et al. indicated that mixed starter culture with Lactobacillus sakei and Staphylococcus significantly reduced putrescine accumulation [45]. Nie et al. found similar results [46].

As shown in Figure 6C, after salting and fermentation, compared with the other four groups, the lowest content of cadaverine (CAD) in the BL4-8 group was 0.641 mg/kg, which was significantly lower than that in other groups (*p* < 0.05), indicating that the starter BL4-8 can degrade CAD. As shown in Figure 6D–F, the contents of phenylethylamine (PHE), histamine (HIS), and tyramine (TYR) in the BL4-8, CL4-3, and X3-2B groups were significantly lower than those in the CO group (*p* < 0.05). The concentration of TYR and HIS in the final product from the present study was 4.10–10.21 mg/kg and 3.79–6.76 mg/kg, respectively, slightly lower than the TYR and HIS concentration in dry fermented sausage (115.80 ± 15.46 mg/kg and 16.55 ± 2.33 mg/kg) reported by Suvajdzic et al. [47]. After high-temperature roasting, the total BAs in samples were less than 500 mg/kg for the CO, PL-4, BL4-8, CL4-3, and X3-2B groups (Figure 6G). Additionally, the total amount of BAs was lower in BL4-8, CL4-3, X3-2B and CO group than the biolamine-positive control group (PL-4). The total biogenic amine content in fermented jerky significantly increased during the roasting period (*p* < 0.05), and in the ripened product this amounted to 103.39–445.29 mg/kg. Wang et al. reported that similar results [23]. The addition of starter BL4-8, CL4-3, and X3-2B reduced the BAs content in fermented lamb jerky. Poisoning can occur when 100 mg of histamine or 10–100 mg of tyramine is consumed [48]. EFSA proposed values for Acute Reference Doses (ARfD), as well as potential ARfD (per adult per meal) for these doses. A healthy individual can tolerate up to 50 mg of HIM, but for individuals with histamine intolerance, only food with levels of this amine below a detectable limit is considered safe. A healthy individual can take 600 mg of TYR in the absence of MAO inhibitors, someone taking MAO inhibitors of the third generation should take less than 50 mg, and someone taking a classical MAO inhibitor should take less than 6 mg. Amounts for PUT and CAD are not reported due to insufficient information [49]; nevertheless, they produce nitrosamine carcinogens when they react with nitrite [6].

## 4. Conclusions

Inoculation of the starter cultures (PL-4, BL4-8, CL4-3, and X3-2B) accelerated the acidification and decreased a_w_, moisture content, nitrite residues, and oxidation degree, and improved the color, hardness, chewiness, elasticity, and the type and content of aldehydes, ketones, acids and ester flavor substances in fermented lamb jerky. However, the starter cultures (BL4-8, CL4-3, and X3-2B) inhibited the accumulation of BAs. The accumulation of BAs in samples achieved our expectation for *L. plantarum* BL4-8–, CL4-3–, and X3-2B–reducing BAs. Jerky made with the starter cultures is safe for consumers. Therefore, *L. plantarum* BL4-8, CL4-3, and X3-2B can be considered potential starters.

## Figures and Tables

**Figure 1 foods-11-02057-f001:**
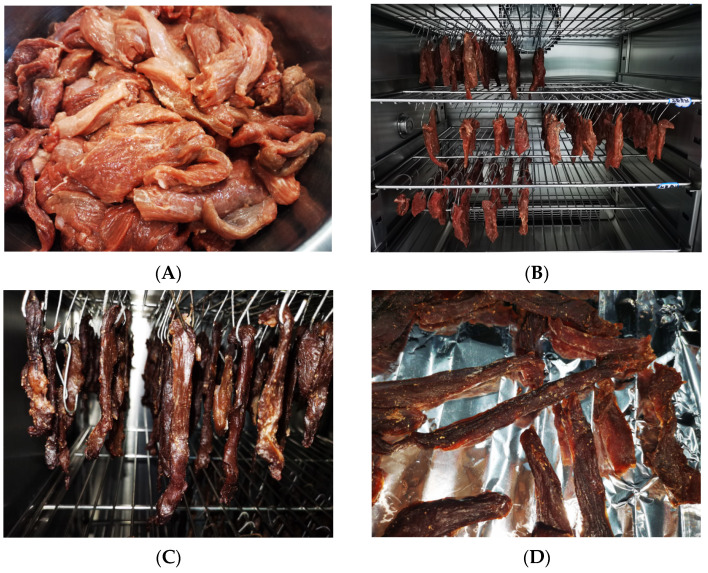
Images of fermented lamb jerky during ripening process ((**A**) Salting; (**B**) Fermentation; (**C**) Low-temperature roasting; (**D**) High-temperature roasting).

**Figure 2 foods-11-02057-f002:**
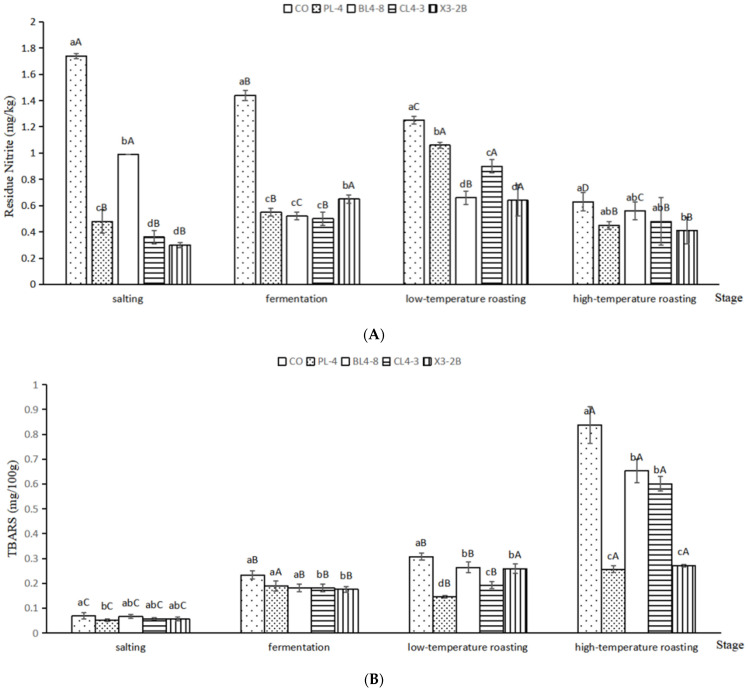
Residue Nitrite (**A**) and TBARS (**B**) in fermented lamb jerky during ripening process. ^A,B,C,D^: Mean values followed different uppercase letters in the same batch indicate significant difference. ^a,b,c,d^: Mean values followed different lowercase letters in the same days of ripening indicate significant difference (*p* < 0.05). CO: control without starter culture; PL-4: with PL-4 *Lactobacillus plantarum*; BL4-8: with BL4-8 *Lactobacillus plantarum*; CL4-3: with CL4-3 *Lactobacillus plantarum*; X3-2B: with X3-2B *Lactobacillus plantarum*.

**Figure 3 foods-11-02057-f003:**
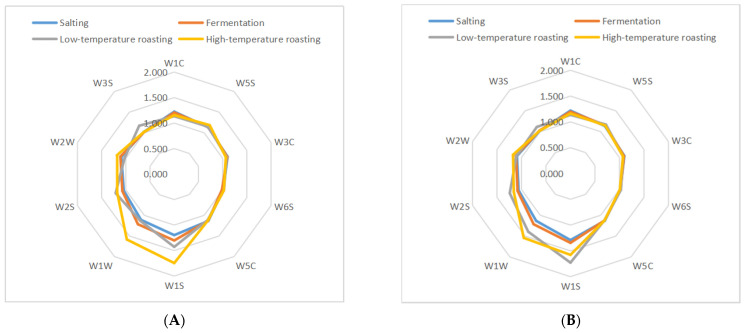
Fingerprint chart of fermented lamb jerky during ripening process ((**A**) CO; (**B**) PL-4; (**C**) BL4-8; (**D**) CL-4; (**E**) X3-2B).

**Figure 4 foods-11-02057-f004:**
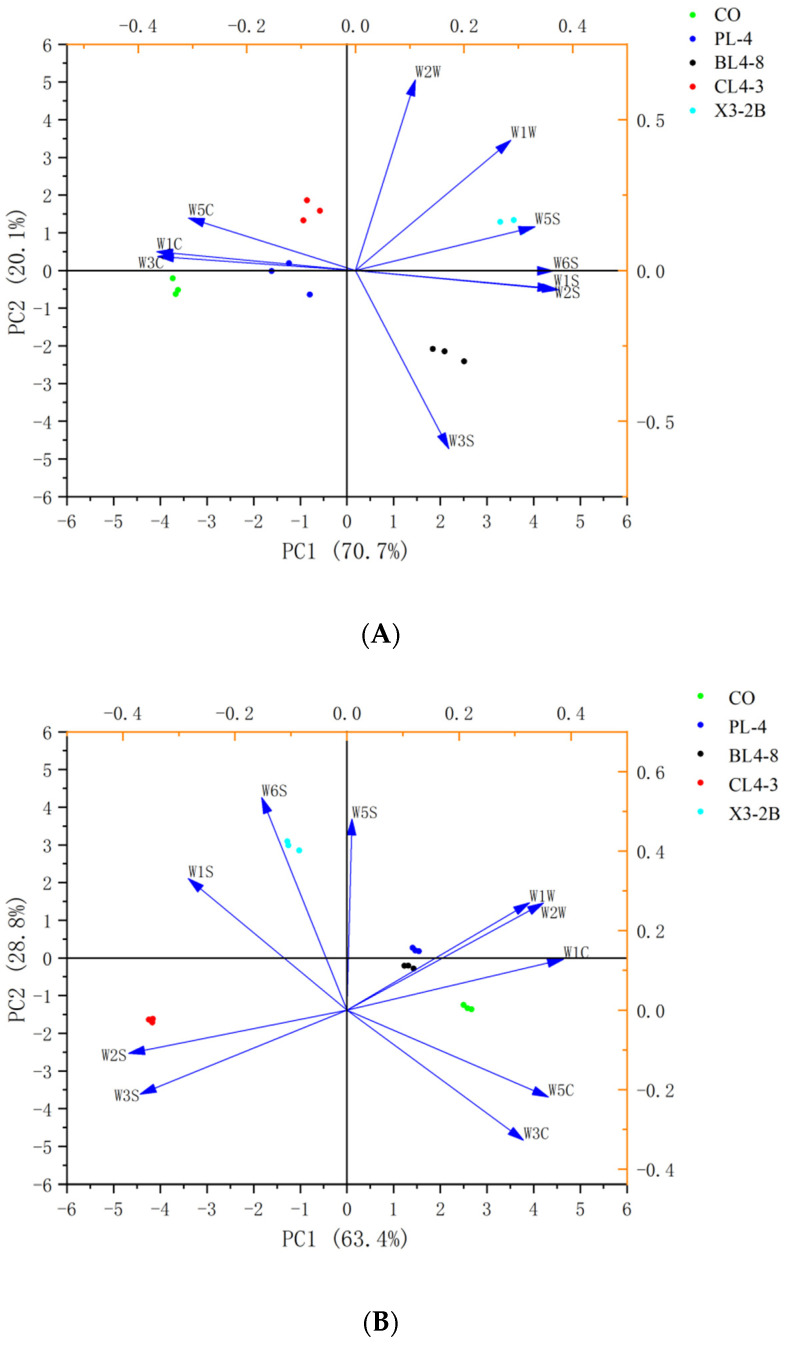
PCA of fermented lamb jerky during ripening process ((**A**) Salting; (**B**) Fermentation; (**C**) Low-temperature roasting; (**D**) High-temperature roasting).

**Figure 5 foods-11-02057-f005:**
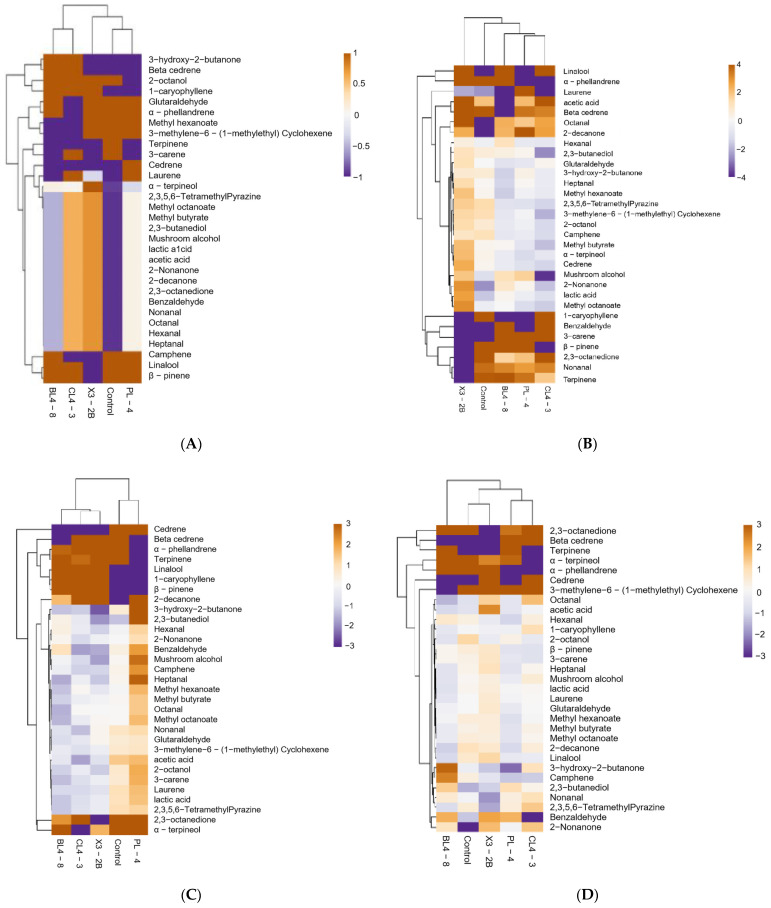
Heatmap of fermented lamb jerky during ripening process ((**A**) Salting; (**B**) Fermentation; (**C**) Low-temperature roasting; (**D**) High-temperature roasting).

**Figure 6 foods-11-02057-f006:**
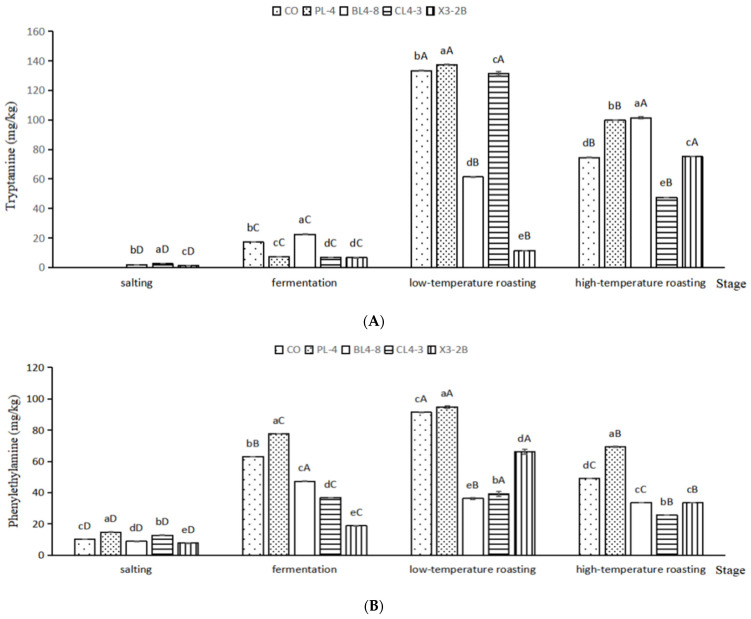
Tryptamine (**A**), putrescine (**B**), cadaverine (**C**), phenylethylamine (**D**), histamine (**E**), tyramine (**F**) and the total BAs content (**G**) in fermented lamb jerky during ripening process. ^A,B,C,D^: Mean values followed different uppercase letter in the same column indicate significant difference. ^a,b,c,d,e^: Mean values followed different lowercase letters in the same row indicate significant difference (*p* < 0.05).

**Table 1 foods-11-02057-t001:** pH, a_w_ and Moisture Content in fermented lamb jerky during ripening process.

	Stage	CO	PL-4	BL4-8	CL4-3	X3-2B
pH	Salting	5.71 ± 0.000 ^aA^	5.46 ± 0.000 ^bB^	5.47 ± 0.000 ^bA^	5.43 ± 0.012 ^cA^	5.41 ± 0.006 ^dB^
	Fermentation	5.22 ± 0.006 ^aD^	5.18 ± 0.006 ^bD^	4.75 ± 0.006 ^eD^	4.95 ± 0.006 ^dD^	5.11 ± 0.006 ^cD^
	Low-temperature roasting	5.46 ± 0.026 ^aC^	5.39 ± 0.006 ^bC^	4.95 ± 0.010 ^dC^	5.02 ± 0.006 ^cC^	5.39 ± 0.010 ^bC^
	High-temperature roasting	5.49 ± 0.006 ^bB^	5.45 ± 0.006 ^aA^	5.00 ± 0.012 ^eB^	5.22 ± 0.006 ^dB^	5.46 ± 0.006 ^cA^
a_w_	Salting	0.863 ± 0.008 ^aA^	0.863 ± 0.003 ^aA^	0.859 ± 0.001 ^aA^	0.876 ± 0.010 ^aA^	0.866 ± 0.010 ^aA^
	Fermentation	0.698 ± 0.001 ^eB^	0.728 ± 0.005 ^cB^	0.743 ± 0.002 ^bB^	0.760 ± 0.003 ^aB^	0.716 ± 0.002 ^dB^
	Low-temperature roasting	0.681 ± 0.004 ^dC^	0.703 ± 0.003 ^aB^	0.668 ± 0.003 ^bC^	0.578 ± 0.000 ^aC^	0.607 ± 0.002 ^cC^
	High-temperature roasting	0.584 ± 0.002 ^dD^	0.547 ± 0.002 ^aC^	0.564 ± 0.002 ^cD^	0.569 ± 0.001 ^bD^	0.547 ± 0.002 ^dD^
Moisture	Salting	0.717 ± 0.002 ^cdA^	0.723 ± 0.003 ^bcA^	0.731 ± 0.000 ^abA^	0.734 ± 0.001 ^aA^	0.709 ± 0.005 ^dA^
content	Fermentation	0.348 ± 0.001 ^eB^	0.393 ± 0.001 ^cB^	0.436 ± 0.004 ^aB^	0.423 ± 0.002 ^bB^	0.382 ± 0.002 ^dB^
	Low-temperature roasting	0.248 ± 0.001 ^eC^	0.395 ± 0.004 ^aB^	0.332 ± 0.002 ^cC^	0.378 ± 0.001 ^bC^	0.271 ± 0.003 ^dC^
	High-temperature roasting	0.203 ± 0.002 ^dD^	0.250 ± 0.001 ^aC^	0.230 ± 0.000 ^bcD^	0.233 ± 0.003 ^bD^	0.226 ± 0.002 ^cD^

^A,B,C,D^: Mean values followed different uppercase letter in the same column indicate significant difference. ^a,b,c,d^^,e^: Mean values followed different lowercase letters in the same row indicate significant difference (*p* < 0.05). CO: control without starter culture; PL-4: with PL-4 *Lactobacillus plantarum*; BL4-8: with BL4-8 *Lactobacillus plantarum*; CL4-3: with CL4-3 *Lactobacillus plantarum*; X3-2B: with X3-2B *Lactobacillus plantarum*.

**Table 2 foods-11-02057-t002:** Color in fermented sausage during ripening process.

	Stage	CO	PL-4	BL4-8	CL4-3	X3-2B
L*	Salting	31.69 ± 0.10 ^aC^	31.61 ± 0.29 ^aD^	30.07 ± 0.17 ^bD^	29.49 ± 0.11 ^cC^	27.88 ± 0.20 ^dC^
	Fermentation	27.10 ± 0.15 ^cD^	33.67 ± 0.14 ^aC^	30.57 ± 0.13 ^bC^	27.12 ± 0.03 ^cD^	26.96 ± 0.39 ^cD^
	Low-temperature roasting	34.31 ± 0.43 ^cB^	36.27 ± 0.18 ^aB^	36.63 ± 0.07 ^aB^	35.31 ± 0.21 ^bB^	35.51 ± 0.32 ^bB^
	High-temperature roasting	36.59 ± 0.16 ^eA^	39.91 ± 0.30 ^aA^	39.21 ± 0.04 ^bA^	37.20 ± 0.42 ^dA^	38.61 ± 0.24 ^cA^
a*	Salting	9.43 ± 0.06 ^cD^	9.82 ± 0.12 ^cC^	10.52 ± 0.14 ^bC^	10.91 ± 0.36 ^bB^	11.65 ± 0.39 ^aAB^
	Fermentation	10.08 ± 0.11 ^bC^	10.13 ± 0.07 ^bC^	10.29 ± 0.22 ^bC^	8.73 ± 0.17 ^cC^	11.44 ± 0.20 ^aB^
	Low-temperature roasting	10.59 ± 0.09 ^bB^	12.22 ± 0.81 ^aB^	11.42 ± 0.06 ^abB^	10.60 ± 0.54 ^bB^	10.75 ± 0.13 ^bB^
	High-temperature roasting	11.61 ± 0.29 ^cA^	13.58 ± 0.18 ^aA^	13.36 ± 0.41 ^abA^	12.72± 0.55 ^abA^	12.54 ± 0.83 ^bcA^
b*	Salting	6.64 ± 0.10 ^bC^	7.39 ± 0.18 ^aD^	6.61 ± 0.06 ^bD^	7.58 ± 0.04 ^aC^	6.63 ± 0.03 ^bC^
	Fermentation	6.40 ± 0.01 ^cD^	7.82 ± 0.08 ^aC^	6.83 ± 0.05 ^bC^	5.36 ± 0.05 ^eD^	6.12 ± 0.12 ^dD^
	Low-temperature roasting	8.08 ± 0.13 ^bB^	8.56 ± 0.34 ^aB^	8.54 ± 0.11 ^aB^	7.97 ± 0.14 ^bB^	8.47 ± 0.08 ^aB^
	High-temperature roasting	11.26 ± 0.11 ^cA^	11.54 ± 0.02 ^bA^	12.12 ± 0.12 ^aA^	11.56 ± 0.20 ^bA^	10.55 ± 0.06 ^dA^
e	Salting	1.72 ± 0.03 ^dB^	1.64 ± 0.02 ^dB^	1.94 ± 0.01 ^bA^	1.81 ± 0.05 ^cB^	2.18 ± 0.07 ^aA^
	Fermentation	1.75 ± 0.02 ^bA^	1.60 ± 0.02 ^dB^	1.84 ± 0.05 ^cB^	1.95 ± 0.05 ^bA^	2.30 ± 0.05 ^aA^
	Low-temperature roasting	1.57 ± 0.03 ^bC^	1.76 ± 0.06 ^aA^	1.65 ± 0.02 ^bC^	1.63 ± 0.09 ^bC^	1.62 ± 0.02 ^bB^
	High-temperature roasting	1.35 ± 0.02 ^bD^	1.52 ± 0.02 ^aC^	1.44 ± 0.03 ^abD^	1.44 ± 0.06 ^abD^	1.51 ± 0.09 ^aB^

^A,B,C,D^: Mean values followed different uppercase letter in the same column indicate significant difference. ^a,b,c,d^^,e^: Mean values followed different lowercase letters in the same row indicate significant difference (*p* < 0.05). CO: control without starter culture; PL-4: with PL-4 *Lactobacillus plantarum*; BL4-8: with BL4-8 *Lactobacillus plantarum*; CL4-3: with CL4-3 *Lactobacillus plantarum*; X3-2B: with X3-2B *Lactobacillus plantarum*.

**Table 3 foods-11-02057-t003:** Texture of fermented lamb jerky.

Texture	CO	PL-4	BL4-8	CL4-3	X3-2B
Hardness (N)	4902.20 ± 228.17 ^a^	1760.96 ± 141.06 ^c^	2099.46 ± 122.68 ^bc^	1041.18 ± 170.98 ^bc^	3513.59 ± 332.05 ^b^
Elasticity	0.58 ± 0.08 ^bc^	0.73 ± 0.07 ^ab^	0.71 ± 0.11 ^ab^	0.53 ± 0.09 ^c^	0.67 ± 0.11 ^b^
Cohesion	0.52 ± 0.12 ^a^	0.75 ± 0.01 ^ab^	0.3 ± 0.17 ^b^	0.63 ± 0.19 ^b^	0.61 ± 0.10 ^b^
Adhesiveness	2540.94 ± 374.51 ^c^	1320.89 ± 133.38 ^b^	1315.19 ± 206.34 ^b^	659.50 ± 29.82 ^a^	2154.47 ± 340.76 ^ab^
Chewiness (N)	1474.25 ± 199.40 ^a^	959.22 ± 90.49 ^b^	926.72 ± 83.30 ^b^	349.46 ± 23.12 ^c^	1326.12 ± 160.80 ^ab^
Resilience	0.16 ± 0.11 ^c^	0.26 ± 0.06 ^a^	0.20 ± 0.09 ^bc^	0.23 ± 0.12 ^ab^	0.20 ± 0.11 ^bc^

^a,b,c^: Mean values followed different lowercase letters in the same row indicate significant difference (*p* < 0.05). CO: control without starter culture; PL-4: with PL-4 *Lactobacillus plantarum*; BL4-8: with BL4-8 *Lactobacillus plantarum*; CL4-3: with CL4-3 *Lactobacillus plantarum*; X3-2B: with X3-2B *Lactobacillus plantarum*.

**Table 4 foods-11-02057-t004:** Odor activity values of key volatile compounds of fermented lamb jerky.

Compound	Threshold (µg/kg in Water)	OAV				
		CO	PL-4	BL4-8	CL4-3	X3-2B
Hexanal	5	157.8	184.2	651	389.4	148.4
Heptanaldehyde	3	71	116.33	126	152.67	178.67
Octanal	3.4	57.65	195	118.82	652.35	366.18
Glutaraldehyde	22	2.59	5.68	7.73	5.91	6.5
Nonanal	1	256	1126	1358	1255	116
Benzaldehyde	320	0.66	16.4	18.89	0.18	10.17
2-decanone	7.94	22.92	19.4	18.89	39.67	30.6
2-Nonanone	200	0.02	0.59	1.29	1.35	1.06
1-octene-3-ol	1	325	698	617	987	635
Linalool	6	12.17	16.33	12.67	13.83	27.33
Methyl butyrate	15.1	153.91	242.25	303.91	345.89	279.8
Methyl octanoate	13	182.23	248.08	319.85	328.38	273.54
Camphene	17	19	18.47	265.53	14.71	14.76
1-caryophyllene	64	0	4.03	10.84	9.2	19.31
β-Pinene	140	1.83	2.56	4.99	1.84	3.47
Methyl caproate	70	46.54	59.79	103.37	73.03	74.8

## Data Availability

The data that are presented in this study are available on request from the corresponding author.

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
