# Peer review of "Effect of Biogenic Amine-Degrading Lactobacillus on the Biogenic Amines and Quality in Fermented Lamb Jerky"

_foods, 2022, doi:10.3390/foods11142057_

Round 1

Reviewer 1 Report

The presented study is quite complete. The topic is very interesting and important to the field. The analytical work is huge and well covered, however needs some improvements to precisely clarify the measurement range of the applied methods, better clarify the analyte determined in certain parts of the manuscript.. The manuscript is complete, the discussion is good. Line 93 Define the measurement units for TBARS and nitrite

Line 94 Describe precisely what the nitrite analysis refers to

Line 96 Volatile flavor Could you add the thresholds of the major flavor substances

Line116 Biogenic Amines

Instrumental equipment description missing, the chromatographic column description as well. In general clarify the major method steps, i.e. the amount of sample used similar to chapter 96 Volatile flavor under line 97. BA concentrations are expressed in XX units – define the measurement units and in general better describe the quantification part.

Line 118 It is not clear what is determined for biogenic amine content. What about the sample, you write about strain solution.

Line 121 and line 124 are confusing. The compounds were quantified using internal or external calibration curves?

Line 155  The concentration levels ranging from 1-50 mg/l. What about the measuring range expressed in mg/kg you later discuss under the 3 6 Biogenic Amines Analysis chapter as well the limit of detection and limit of quantification

Line 188 Nitrite is not precise. Clarify what do you measure and the analyte you express (is it NO2- ion or sodium nitrite you added in fermented lam jerky formulation

Line 311 Specify the detector used and move the instrument description under the Biogenic Amine chapter

Line 344 Figure 6 description is missing

Author Response

Dear Reviewer:

Thank you for your comments and suggestions on our manuscript. The comments and suggestions are valuable for improving our paper. We have read the comments carefully and revised accordingly. Any revisions to the manuscript have be marked up using the “Track Changes” function.

We hope our revised version will now be acceptable for publication in Foods. Thank you for your time and consideration.

Best regards.

Response to Reviewer 1 Comments

Point 1: The manuscript is complete, the discussion is good. Line 93 Define the measurement units for TBARS and nitrite

Response 1: Thank you very much for your comment. We have modified introduction according to your suggestion. The measurement units for TBARS and nitrite are mg/100g and mg/kg respectively. Please see page 3, line 101.

Point 2: Line 94 Describe precisely what the nitrite analysis refers to

Response 2: Thank you for your comment. We have modified introduction according to your suggestion. The nitrite analysis means residue nitrite analysis. Please see page 3, line 101.

Point 3: Line 96 Volatile flavor Could you add the thresholds of the major flavor substances

Response 3: Thank you for your comment. We very much agree to this point and changed our manuscript accordingly. We added the the thresholds of the major flavor substances. Please see page 3, line 115-117 and Table 4.

Point 4: Line116 Biogenic Amines

Instrumental equipment description missing, the chromatographic column description as well. In general clarify the major method steps, i.e. the amount of sample used similar to chapter 96 Volatile flavor under line 97. BA concentrations are expressed in XX units – define the measurement units and in general better describe the quantification part.

Response 4: Thank you for your comment. We have modified introduction according to your suggestion. Please see page 3, line 130-137.

Point 5: Line 118 It is not clear what is determined for biogenic amine content. What about the sample, you write about strain solution.

Response 5: Thank you for your comment. We have modified according to your suggestion. Please see page 3, line 129-141.

Point 6: Line 121 and line 124 are confusing. The compounds were quantified using internal or external calibration curves?

Response 6: Thank you for your comment. We have modified according to your suggestion. The compounds were quantified using internal calibration curves. Please see page 3, line 137-141.

Point 7: Line 155  The concentration levels ranging from 1-50 mg/l. What about the measuring range expressed in mg/kg you later discuss under the 3 6 Biogenic Amines Analysis chapter as well the limit of detection and limit of quantification

Response 7: Thank you for your comment. We have modified according to your suggestion. Please see page 12, line 391-395.

Point 8: Line 188 Nitrite is not precise. Clarify what do you measure and the analyte you express (is it NO2- ion or sodium nitrite you added in fermented lam jerky formulation

Response 8: Thank you for your comment. We added sodium nitrite in fermented lamb jerky formulation. The nitrite means residue nitrite. Please see page 6, line 211-214.

Point 9: Line 311 Specify the detector used and move the instrument description under the Biogenic Amine chapter

Response 9: Thank you for your comment. We have modified according to your suggestion. Please see page 3, line 134-137.

Point 10: Line 344 Figure 6 description is missing

Response 10: Thank you for your comment. We have added description of Figure 6. Please see page 14, line 470-472.

Reviewer 2 Report

In this work five groups of sausages were compared for the evolution (during fermentation and ripening steps) of physical-chemical quality, flavor, and biogenic amines.

The topic is of interest, but English makes it tiring to read the manuscript.

In addition, the data could be better presented, with clearer figures (it should be not only improved the resolution of figures, but data elaboration too, it should be carefully checked the statistical analysis).

In particular, my suggestions are below:

 Methods

The methods used for texture need to be described.

Results

Line 131 (and all text and figures): In the text the abbreviation of "water activity" must be corrected (correct: aw).

Figure 1: it is not clear how it results statistically significant differences for these data. Please carefully check statistical analysis. Maybe better a table?

Table 1. it is not clear how it results statistically significant differences for these data. Please carefully check statistical analysis.

Table 2 also does not specify the unit of measurement for the quantities considered.

Figure 3: It should be better to show the Fingerprint chart of fermented lamb jerky during process for each groups, not all for each step. Moreover, the figure should put in evidence statistically significant differences.

Figure 6: it is illegible

 Discussion: data are shown but not properly discussed. Above all for the following subsections:

3.4.Electronic Nose Analysis: these data are shown but none reference is cited. Please improve the discussion of the data referring to appropriate bibliography

3.5. Volatile Flavor Analysis: in this paragraph only one reference is cited. Please improve the discussion of the data referring to appropriate bibliography

3.5. Volatile Flavor Analysis: in this paragraph only one reference is cited. Please improve the discussion of the data referring to appropriate bibliography

Author Response

Dear Reviewer:

Thank you for your comments and suggestions on our manuscript. The comments and suggestions are valuable for improving our paper. We have read the comments carefully and revised accordingly. Any revisions to the manuscript have be marked up using the “Track Changes” function.

We hope our revised version will now be acceptable for publication in Foods. Thank you for your time and consideration.

Best regards.

Response to Reviewer 2 Comments

Point 1: The methods used for texture need to be described.

Response 1: Thank you for your comment. We have modified according to your suggestion. Please see page 3, line 99-100.

Point 2: Line 131 (and all text and figures): In the text the abbreviation of "water activity" must be corrected (correct: aw).

Response 2: Thank you for your comment. We have modified according to your suggestion in all text and figures.

Point 3: Figure 1: it is not clear how it results statistically significant differences for these data. Please carefully check statistical analysis. Maybe better a table?

Response 3: Thank you for your comment. We have modified according to your suggestion. Please see Table 1.

Point 4: Table 1. it is not clear how it results statistically significant differences for these data. Please carefully check statistical analysis.

Response 4: Thank you for your comment. We have modified according to your suggestion. Please see Table 2.

Point 5: Table 2 also does not specify the unit of measurement for the quantities considered.

Response 5: Thank you for your comment. Thank you for your comment. We have modified according to your suggestion. Please see Table 3.

Point 6: Figure 3: It should be better to show the Fingerprint chart of fermented lamb jerky during process for each groups, not all for each step. Moreover, the figure should put in evidence statistically significant differences.

Response 6: Thank you for your comment. We have modified according to your suggestion. Please see Figure 2. We added the Table S1 put in evidence statistically significant differences. Please see page 17-18, line 590-594.

Point 7: Figure 6: it is illegible

Response 7: Thank you for your comment. We have modified according to your suggestion. Please see Figure 5.

Point 8: Discussion: data are shown but not properly discussed. Above all for the following subsections:

Response 8: Thank you for your comment. We have modified according to your suggestion.

Point 9: 3.4.Electronic Nose Analysis: these data are shown but none reference is cited. Please improve the discussion of the data referring to appropriate bibliography

Response 9: Thank you for your comment. We have modified according to your suggestion. Please see page 7-8, line 250-307.

Point 10: 3.5. Volatile Flavor Analysis: in this paragraph only one reference is cited. Please improve the discussion of the data referring to appropriate bibliography

Response 10: Thank you for your comment. We have modified according to your suggestion. Please see page 9-10, line 323-386.

Reviewer 3 Report

The article describes the effect of various strains of Lactobacillus plantarum as potential starter cultures in the product called "Jerky", sausage made from lamb meat. It is a product that, although it undergoes a brief initial fermentation process, undergoes two subsequent heat treatments, one of which at 150ºC.

It is a very particular product for which it is not clear whether it is a product for very local consumption or one that is widely distributed (no data is provided on its level of consumption, it is only mentioned that it is a product “welcomed and loved by the majority of consumers”, but without giving a clear idea of ​​which consumers it refers to), so the interest of the work is relative.

The effect of the addition of the 4 strains studied on different quality parameters, such as pH, Aw, color, residual nitrites, TBARS, texture and volatile compounds, is assessed. Biogenic amines are only a small part of the work, although they are highlighted both in the title and in the final conclusions.

The "positive" role of some of the strains studied in the quality parameters of the product is highlighted, but a sensory analysis is not included to confirm whether these changes actually imply a better evaluation by the consumer.

The article presents writing that is difficult to follow in some sections, especially in materials and methods.

Other minor considerations

Ln 25-26: Provide data that support this statement. What do you mean by "most consumers"? What geographical area does its consumption extend to?

Ln 31-33: Is this an industrial product, or is it a homemade product?

Ln 41. Is the role of BA as precursors of nitrosamines fully demonstrated? It appears to be based on a secondary reference; include the original reference to support this claim.

Ln 53, 75, 352, 353: write the name of the microorganisms in italics. Ln 70 correct the position of the “s”.

Ln 78: write in superscript

Ln 84: is “baking” the correct term for this process?

Ln 94: Was the TBARS analysis actually carried out by these authors, or does it mean that the method they describe was used?

Ln 136: What are "proteolytic amines"?. BAs are not formed by the action of proteases alone, but rather the action of decarboxylases is required.

Ln. 157: What's “e” value? Do you mean “delta E”?

Use of decimals throughout the manuscript: the use of 3 decimal places is unnecessary and causes some confusion.

Ln. 339 . What are biolamines?

340-342: The reference used for this statement does not seem appropriate. Legislated levels for some biogenic amines (histamine) in the US and EU are much lower than this figure. In addition, it must be taken into account that the level of individual sensitivity has a great influence on the harmful effects of AB.

The figure showing the BA levels has no title. The columns of the graphs that show the levels of BA do not show the variability and do not allow to see to what extent the reduction in the formation of the BA by the use of the starter cultures is significant with respect to the CO samples.

Author Response

Dear Reviewer:

Thank you for your comments and suggestions on our manuscript. The comments and suggestions are valuable for improving our paper. We have read the comments carefully and revised accordingly. Any revisions to the manuscript have be marked up using the “Track Changes” function.

We hope our revised version will now be acceptable for publication in Foods. Thank you for your time and consideration.

Best regards.

Response to Reviewer 3 Comments

Point 1: The article describes the effect of various strains of Lactobacillus plantarum as potential starter cultures in the product called "Jerky", sausage made from lamb meat. It is a product that, although it undergoes a brief initial fermentation process, undergoes two subsequent heat treatments, one of which at 150ºC.

Response 1: Thank you for your comment. We used Lactobacillus plantarum as potential starter cultures in the product, undergoes two subsequent heat treatments were carried out according to the process of the factory, the first heat treatments is for dehydration, and the second heat treatment is to make it into finished products.

Point 2: It is a very particular product for which it is not clear whether it is a product for very local consumption or one that is widely distributed (no data is provided on its level of consumption, it is only mentioned that it is a product “welcomed and loved by the majority of consumers”, but without giving a clear idea of ​​which consumers it refers to), so the interest of the work is relative.

Response 2: Thank you for your comment. We added the survey showed that 88.0% of consumers have eaten leisure jerky products. Please see page 1, line 30.

Point 3: The effect of the addition of the 4 strains studied on different quality parameters, such as pH, Aw, color, residual nitrites, TBARS, texture and volatile compounds, is assessed. Biogenic amines are only a small part of the work, although they are highlighted both in the title and in the final conclusions.

Response 3: Thank you for your comment. This study compared five types of lamb jerky, namely, CO (without starter culture), PL-4 (with producing putrescine, cadaverine, histamine, and tyramine), BL4-8 (degrading putrescine, cadaverine, histamine, and tyramine), CL4-3 (degrading putrescine and tyramine), and X3-2B (degrading histamine and tyramine). The levels of histamine, putrescine, and tyramine were significantly lower in the BL4-8, CL4-3, and X3-2B groups than in CO and PL-4 groups. So biogenic amines are highlighted both in the title and in the final conclusions.

Point 4: The "positive" role of some of the strains studied in the quality parameters of the product is highlighted, but a sensory analysis is not included to confirm whether these changes actually imply a better evaluation by the consumer.

Response 4: Thank you for your comment. We understand that sensory analysis may better reveal the these changes actually imply a better evaluation by the consumer. However, in the present study, we mainly focused on the different quality parameters, such as color, residual nitrites, TBARS, texture and volatile compounds, and we think that may not be optimal, but should be sufficient to draw a conclusion that the "positive" role of some of the strains studied in the quality parameters of the product. In addition, due to the COVID-19 and government control, we cannot enter the laboratory to supplement the experiment

Point 5: The article presents writing that is difficult to follow in some sections, especially in materials and methods.

Response 5: Thank you for your comment. We have modified materials and methods according to your suggestion. Please see page 2-3, line 73-145.

Point 6: Ln 25-26: Provide data that support this statement. What do you mean by "most consumers"? What geographical area does its consumption extend to?

Response 6: Thank you for your comment. The main geographical area does its consumption extend to Inner Mongolia, Liaoning, Sichuan, and Henan in China. We have modified according to your suggestion. Please see page 1, line 29.

Point 7: Ln 31-33: Is this an industrial product, or is it a homemade product?

Response 7: Thank you for your comment. This is a homemade product.

Point 8: Ln 41. Is the role of BA as precursors of nitrosamines fully demonstrated? It appears to be based on a secondary reference; include the original reference to support this claim.

Response 8: Thank you for your comment. Domestic animals fed nitrite-preserved herring meal developed severe liver disorders which were attributed to the presence of N-nitrosamine as a result of the reaction between the endogenous amines of the fish meal and the added nitrite. We refered to the literature, Naturwissenschaften, 1964, 51(24), 637-638.

Point 9: Ln 53, 75, 352, 353: write the name of the microorganisms in italics. Ln 70 correct the position of the “s”.

Response 9: Thank you for your comment. We have modified according to your suggestion. Please see page 2 and 15, line 59, 62, 76, 81, 479 and 481.

Point 10: Ln 78: write in superscript

Response 10: Thank you for your comment. We have modified according to your suggestion. Please see page 2, line 85.

Point 11: Ln 84: is “baking” the correct term for this process?

Response 11: Thank you for your comment. We apologize for the error. We have revised it to roasting, and we refered to the literature, Foods, 2021, 10, 2676; Food Chemistry, 2022, 385, 132629-132629; Food Control, 2022, 138, 109038.

Point 12: Ln 94: Was the TBARS analysis actually carried out by these authors, or does it mean that the method they describe was used?

Response 12: Thank you for your comment. It mean that the method they describe was used. We have modified according to your suggestion. Please see page 3, line 101-102.

Point 13: Ln 136: What are "proteolytic amines"?. BAs are not formed by the action of proteases alone, but rather the action of decarboxylases is required.

Response 13: Thank you for your comment. Bacterial proteases induce proteolytic degradation, generating peptides, amino acids, and amines which have a buffering effect on the organic acids produced by lactic acid bacteria during fermentation. We refered to the literature, Food Microbiol, 2011, 28(5), 839-847; Foods, 2021, 10, 2939.

Point 14: Ln. 157: What's “e” value? Do you mean “delta E”?

Response 14: Thank you for your comment. e value, as the comprehensive value of meat color, is calculated by the following formula: e=a*/L+a*/b*. We have modified according to your suggestion. Please see page 3, line 98-99.

Point 15: Use of decimals throughout the manuscript: the use of 3 decimal places is unnecessary and causes some confusion.

Response 15: Thank you for your comment. We apologize for the error. We have revised it to 2 decimal places. 

Point 16: Ln. 339 . What are biolamines?

Response 16: Thank you for your comment. We apologize for the error. We have revised it to BAs. Please see page 13, line 424.

Point 17: 340-342: The reference used for this statement does not seem appropriate. Legislated levels for some biogenic amines (histamine) in the US and EU are much lower than this figure. In addition, it must be taken into account that the level of individual sensitivity has a great influence on the harmful effects of AB.

Response 17: Thank you for your comment. We have modified according to your suggestion. Please see page 13, line 426-429.

Point 18: The figure showing the BA levels has no title. The columns of the graphs that show the levels of BA do not show the variability and do not allow to see to what extent the reduction in the formation of the BA by the use of the starter cultures is significant with respect to the CO samples.

Response 18: Thank you for your comment. We have added title of Figure 5 and We have modified Figure 5 (G) according to your suggestion. Please see page 14, line 470-472 and Figure 5.

Round 2

Reviewer 2 Report

the manuscript has been improved and the suggested revisions have been appropriately made. The low resolution of the figures remains

Author Response

Dear Reviewer:

Thank you for your comments and suggestions on our manuscript. The comments and suggestions are valuable for improving our paper. We have read the comments carefully and revised accordingly. Any revisions to the manuscript have be marked up using the “Track Changes” function.

We hope our revised version will now be acceptable for publication in Foods. Thank you for your time and consideration.

Best regards.

Response to Reviewer 2 Comments

Point 1: the manuscript has been improved and the suggested revisions have been appropriately made. The low resolution of the figures remains.

Response 1: Thank you for your comment. We have improved the resolution of the figures according to your suggestion. Please see all figures.

Reviewer 3 Report

The modifications introduced by the authors allow some important aspects to be better understood. However, there are still aspects that should be improved:

Jerky is quite a peculiar product, since it includes fermentation and cooking stages. It is not a whole piece, since it is cut, but it is not indicated that it is stuffed into a casing or similar, so it is difficult to compare it with other better-known meat products. It would be interesting to describe the usual preparation procedure (not only the one used in the study, adapted to its objectives), and even include some images of it.

As it is a homemade product, it is difficult to understand what impact the use of starter cultures would have, if they are not used regularly in its preparation. How do the authors think its use could be encouraged?

Answer to point 8. It is clear that the N-nitrosamines use secondary amines as precursors, but not the repercussion of the formation of amines such as HIS, TYR, PUT or CAD. The 1964 reference is somewhat old.

Response to point 13: this explanation should be included in the text, but in any case decarboxylases would be necessary.

The use of English should be reviewed: Ln 119 "compriseding"

Tables 1 and 2: separate with a horizontal line the results of pH, Aw and moisture (table 1) and the values ​​L, a, b and 3 (table 2).

Ln 391-394: this text corresponds more to the M&M section

Point 3.6. Some discussion is missing comparing the BA levels found in Jerky and those reported in other meat products (eg Ln 46-52).

Response to point 17: Ln 427-429: the interpretation of the level of risk is too simple: I recommend a more detailed reading of document 46, which provides a lot of data in this regard.

In figure 5 the lines that indicate the variability (SD or SE) are still missing, in some cases the differences described in the text do not seem so evident.

Author Response

Dear Reviewer:

Thank you for your comments and suggestions on our manuscript. The comments and suggestions are valuable for improving our paper. We have read the comments carefully and revised accordingly. Any revisions to the manuscript have been marked up using the “Track Changes” function.

We hope our revised version will now be acceptable for publication in Foods. Thank you for your time and consideration.

Best regards.

Response to Reviewer 3 Comments

Point 1: Jerky is quite a peculiar product, since it includes fermentation and cooking stages. It is not a whole piece, since it is cut, but it is not indicated that it is stuffed into a casing or similar, so it is difficult to compare it with other better-known meat products. It would be interesting to describe the usual preparation procedure (not only the one used in the study, adapted to its objectives), and even include some images of it.

Response 1: Thank you for your comment. We have been supplemented the usual preparation procedure and some images of fermented jerky according to your suggestion. Please see page 2, line 85-93 and Figure 1.

Point 2: As it is a homemade product, it is difficult to understand what impact the use of starter cultures would have, if they are not used regularly in its preparation. How do the authors think its use could be encouraged?

Response 2: We use injection method to add the starter to the upper, middle and lower parts of the meat strip to ensure the regularly of the starter, facilitate the regularly of the product and ensure its quality. We refered to the literature about injection method, Journal of Food Processing and Preservation, 2021;45:e15744.

Point 3: Answer to point 8. It is clear that the N-nitrosamines use secondary amines as precursors, but not the repercussion of the formation of amines such as HIS, TYR, PUT or CAD. The 1964 reference is somewhat old.

Response 3: Thank you for your comment. We have modified the literature (Food Chemistry, 2011, 126(4), 1539-1545) in text according to your suggestion. Please see page 19, line 541-542. The research showed that spermidine and putrescine amplify the formation of N-nitrosodimethylamine (NDMA), spermidine and cadeverine cause a significant increase of N-nitrosopiperidine (NPIP) (Food Chemistry, 2011, 126(4), 1539-1545). This view is also mentioned in the literature, EFSA Journal 2011;9(10):2393. In addition, the formation of N-nitrosopyrrolidine (NPYR) and NPIP from putrescine, spermidine and cadaverine was studied in a model experiment, using minced bacon as the reaction medium (IARC Scientific Publications, 1980, 31: 183-193). Primary amines such as putrescine and cadaverine have been suggested to cyclize during heating to secondary amines such as pyrrolidine and piperidine, which react with nitrite to form carcinogenic nitrosamines (Critical Reviews in Food Science and Nutrition, 2009, 49: 369-377; J. Neuroscience Res. 1983, 9:279–289; J Agric. Food chem. 1975, 23:898–902).

Point 4: Response to point 13: this explanation should be included in the text, but in any case decarboxylases would be necessary.

Response 4: Thank you for your comment. We quite agree with you, and we have modified according to your suggestion. Please see page 4, line 182-186.

Point 5: The use of English should be reviewed: Ln 119 "compriseding"

Response 5: Thank you for your comment. We apologize for the error. We have revised it to comprising. Please see page 4, line 145.

Point 6: Tables 1 and 2: separate with a horizontal line the results of pH, Aw and moisture (table 1) and the values ​​L, a, b and 3 (table 2).

Response 6: Thank you for your comment. We have modified according to your suggestion. Please see Tables 1 and 2.

Point 7: Ln 391-394: this text corresponds more to the M&M section

Response 7: Thank you for your comment. We have modified according to your suggestion. Please see page 4, line 166-169.

Point 8: Point 3.6. Some discussion is missing comparing the BA levels found in Jerky and those reported in other meat products (eg Ln 46-52).

Response 8: Thank you for your comment. We have added the discussion according to your suggestion. Please see page 15 and 16, line 475-477, 483-487 and 490-493.

Point 9: Response to point 17: Ln 427-429: the interpretation of the level of risk is too simple: I recommend a more detailed reading of document 46, which provides a lot of data in this regard.

Response 9: Thank you for your comment. We have modified according to your suggestion. Please see page 16, line 495-503.

Point 10: In figure 5 the lines that indicate the variability (SD or SE) are still missing, in some cases the differences described in the text do not seem so evident.

Response 10: Thank you for your comment. We have modified according to your suggestion. Please see page 18, line 507-509. It showed that BL4-8, CL4-3 and X3-2B strains significantly reduced the content of putrescine, histamine, and tyramine (p<0.05), the total amount of BAs was lower in BL4-8, CL4-3, X3-2B and CO group than the biolamine-positive control group (PL-4) in paper.

Round 3

Reviewer 3 Report

The manuscript has been considerably improved and the answers to the questions have been satisfactory.